# BlackLivesMatter in Healthcare: Racism and Implications for Health Inequity among Aboriginal and Torres Strait Islander Peoples in Australia

**DOI:** 10.3390/ijerph18094399

**Published:** 2021-04-21

**Authors:** Kathomi Gatwiri, Darlene Rotumah, Elizabeth Rix

**Affiliations:** 1Center for Children & Young People, Faculty of Health, Southern Cross University, Gold Coast, QLD 4225, Australia; 2Gnibi College, Southern Cross University, Gold Coast, QLD 4225, Australia; darlene.rotumah@scu.edu.au; 3Faculty of Health, Southern Cross University, Gold Coast, QLD 4225, Australia; liz.rix@scu.edu.au

**Keywords:** racism, aboriginal and Torres Strait Islander peoples, first nations peoples, social determinants of health, cultural safety, decolonization, Australia, yarning, healthcare, whiteness

## Abstract

Despite decades of evidence showing that institutional and interpersonal racism serve as significant barriers to accessible healthcare for Aboriginal and Torres Strait Islander Peoples, attempts to address this systemic problem still fall short. The social determinants of health are particularly poignant given the socio-political-economic history of invasion, colonisation, and subsequent entrenchment of racialised practices in the Australian healthcare landscape. Embedded within Euro-centric, bio-medical discourses, Western dominated healthcare processes can erase significant cultural and historical contexts and unwittingly reproduce unsafe practices. Put simply, if Black lives matter in healthcare, why do Aboriginal and Torres Strait Islander Peoples die younger and experience ‘epidemic’ levels of chronic diseases as compared to white Australians? To answer this, we utilise critical race perspectives to theorise this gap and to de-center whiteness as the normalised position of ‘doing’ healthcare. We draw on our diverse knowledges through a decolonised approach to promote a theoretical discussion that we contend can inform alternative ways of knowing, being, and doing in healthcare practice in Australia.

## 1. A Brief Introduction to Institutionalised Racism in Healthcare in Australia

We start this paper with the story of Naomi Williams, an Aboriginal woman from the Wiradjuri nation. Naomi was 27 years old and 6 months pregnant with her first child when she died of septicaemia in 2016. Her death followed multiple failed attempts to seek treatment at her local hospital. Naomi’s family waited over three years for the findings of an inquest into her death. The inquest heard that Naomi made 18 visits to Tumut hospital in the months before she and her unborn son died, without receiving a referral to any specialist. Instead, nurses assumed she was drug-seeking and lying about her level of pain despite multiple presentations with nausea, vomiting, dehydration, and abdominal pain [1]. On her final presentation to the hospital, Naomi was given two paracetamol tablets and sent home with no physical examination. She died 15 h later of meningococcal septicemia. The inquest heard that Naomi had lowered expectations of receiving dignified care. Prior to her death, she had confided to family and friends that she did not want to go to hospital because “they treated me like a junkie”. There was a finding of racial bias by hospital staff which led to misdiagnosis and the ultimate death of Naomi. Without any reference to her hospital notes, nurses missed clear signs that Naomi had a high-risk pregnancy. Naomi’s fear and consequent death confirms what is already shown in existing literature: The ongoing systemic and individual racism in health care systems significantly informs Aboriginal and Torres Strait Islanders Peoples health seeking decisions [2,3,4].

While there exists a dominant political ideology that positions multiculturalism as evidence of a post-racial Australia, strong evidence supported by the lived experiences of Black people and people of colour in Australia does not support this “race-free” narrative [5,6,7,8,9]. During the re-emergence of the BlackLivesMatter protests globally, conversations about race and racism in Australia were particularly heightened. The #AboriginalLivesMatter movement in the Australian context protested the increasing rates of incarceration and deaths in custody, poor health, and early death for First Nations Peoples in Australia [10]. The movement necessitated conversations on how structures of white supremacy enshrine racism and consequently, contribute to these poor outcomes. To put this in context, it is 30 years since the Royal Commission into Aboriginal and Torres Strait Islander Peoples deaths in custody. Since then, close to 500 deaths in custody have occurred but there has not been a single conviction for those responsible. We question there can there be 500 victims and zero perpetrators. To date, Australia has the highest rates of incarceration of its Indigenous Peoples globally with Aboriginal women being the highest group imprisoned across the country [11,12]. Despite nationwide protest marches in April 2021, Australian mainstream media did not offer much airtime to the issue contributing to the systemic silencing of the institutional killing of Australia’s First Nations Peoples.

Systemic and individual racism have been embedded within Australia’s healthcare delivery since invasion in 1788 [13,14]. After the declaration of Australia as ‘terra nullius’ (land without people), the penal colony was established. First Nation’s Peoples have only been allowed the right to vote since 1962 with no constitutional recognition until a 1967 referendum. There is still no treaty between First Nations Peoples and white Australia, and minimal political representation within the Federal parliament remains the norm [15]. This history, founded in colonial violence, genocide, and the “Australia for the white man” rhetoric informs the ongoing health inequities experienced by First Nations Peoples. The much talked about yet enduring ‘gap’ between the health and well-being of white Australians and First Nations Peoples rests upon this history [16].

Despite increased awareness of the magnitude of the health gap, negligible long-term, culturally safe, and appropriate efforts to remedy this problem have been made [17,18]. Health inequities experienced by Australia’s First Nations Peoples have been theorised and researched for decades [19,20,21]. This evidence however, has had minimal impact on ‘closing the health gap’ and reducing institutional racism [16,22]. In 2017, the then Prime Minister described chronic poor health within Aboriginal and Torres Strait Islander communities as a result of poor “lifestyle choices” and personal failing. This was despite overwhelming evidence that health inequities are directly linked to impacts of colonisation, intergenerational trauma and flawed policies aimed at dealing with what was/is viewed by white Australia as the ‘Aboriginal problem’ [23,24]. Racism and lack of cultural safety as such, remain major barriers to accessible healthcare services for Aboriginal and Torres Strait Islander Peoples [25].

Centering Aboriginal and Torres Strait Islander Peoples’ ways of knowing, being, and doing, this paper shares our combined learnings and applies these to the theoretical approaches of Critical Race Theory, yarning, storytelling, and relational accountability. We aim to amplify calls to prioritise culturally safer healthcare services and institutions in Australia. We utilise our professional and academic lenses to argue that indigenising health practices, and increasing cultural expertise and safety across all health disciplines, is urgent and essential for reducing health inequities for First Nation Peoples in Australia.

## 2. Social Determinants of First Nations Peoples Health

The conditions in which people are born, grow, live, and work are intrinsically linked to their ability to access a good quality of life, which informs their health across the lifespan [26,27]. Popular social determinants of health perspectives often reference poverty as a key determining factor in poor health. Definitions of what poverty is, and who is poor, however, are not always straightforward because the concept of poverty is theoretically complex. Poverty is multi-dimensional, and should not be employed as a singularised explanation of poor health in colonised communities. Defining poverty through Eurocentric material lenses tends to sideline the immense richness of relationships, connection to nature, other human beings, and to ourselves.

First Nation Peoples’ health inequities are profoundly shaped by the lingering violence of colonisation which has informed the entrenchment of institutionalised racism in health systems. Living in a body that is constantly dehumanised through racism increases experiences of racial battle fatigue [28] and racial trauma, both which have negative implications to health [25]. Health determinants for First Nations peoples are informed by choicelessness, cultural, historical, social, and political dimensions [29]. Atkinson [30] was one of the first scholars to fully articulate the direct causal link between colonisation, loss of culture, transgenerational trauma, and a lifetime of increased vulnerability to chronic disease [30,31].

Another factor to consider while applying the SDoH model in understanding First Nations Peoples’ health inequities is the impact of pathologising Aboriginal bodies. Canadian scholar Mary-Ellen Kelm argued in *Colonizing Bodies* that Indigenous peoples are not only physically impacted by colonising policies such as: restrictions to hunting and fishing, the forced removal of children, criminalisation of cultural practices and traditional healing, and loss of language, they are also harmed by the trauma of pathologisation used in humanitarianism and biomedicine to justify why they need to assimilate to white standards of health [32].

### Connection to Country as an SDoH 

In Australia, assimilation policies saw many generations of children forcibly removed from their countries, land, and families in what is known as the *Stolen Generations* [33]. Systemic removal of children from their cultural land saw generations of Aboriginal and Torres Strait Islander peoples experience the erasure of cultural identity. Disconnection from country, culture, language, and family are cited as major contributors to the endemic proportions of chronic disease due to the intergenerational trauma accrued from these histories [21,30,34]. Healing therefore, can transpire through reconnection to country, culture, and community from which they were removed and disconnected [35,36]. Country, culture, and sacred lands are more than just place. They encompass a complex mix of community agency, social connection, and spiritual relationships. When First Nations Peoples move through the country, they are one with it, with emphasis on moving through it, not across it or on top of it [37,38]. This spiritual context is deepened by relationships to ancestral knowledge systems of the country as passed down through Yarning with elders across generations. 

Despite the profound social and emotional trauma inflicted by colonisation, connection to country is intrinsically linked to a wholesome cultural identity and resilience, and is crucial for amelioration of the impacts of colonisation, dispossession, discrimination, loss, grief, and much more [36,39,40,41]. Cultural healing, therefore, is a powerful tool to counter negative impacts on the SDoH, providing connection and identity by increased social and emotional well-being [31].

## 3. Theoretical Articulations: Whiteness and Critical Race Theory in Healthcare

Increasing evidence from health and social research strongly indicates that racism is a public health issue [20,42,43]. Practitioners and medical practices that embody racial prejudices are signalled as contributing to the health seeking behaviours of Aboriginal and Torres Strait Islander Peoples. Due to inaccessibility to culturally safe healthcare services, addressing health inequities requires a closer investigation of how interpersonal racism, i.e., how healthcare practitioners interact with Aboriginal patients, and institutional racism (how systems respond to Aboriginal people’s health) intersect to produce significant health disparities between white Australians and Aboriginal and Torres Strait Islander Peoples. Critical Race Theory (CRT) investigates complex racial processes such as racial hierarchisation and race (un)consciousness allows for the omnipresence of whiteness in healthcare practice in Australia to exist as the norm [20,40]. CRT positions racism as an intersection between prejudice and power and goes beyond individualised cognitive processes of racial stereotyping. In Australia, the power of whiteness is summoned through its assumed default position as the “standard against which differences, or deviations from that norm, are measured” [39], while remaining “invisible, natural, normal, and unmarked” [42]. Whiteness as a symbol of status and social capital dominates other ways of experiencing the world, which consequently undermines Aboriginal and Torres Strait Islander Peoples’ ways of knowing as well as their ontological and axiological ways of doing and being [20,39,42]. Through this unscrutinised invisibility, white Australia continues to accrue institutional power, social, and cultural privilege while First Nations Peoples continue to experience systemic marginalisation.

In Australia, institutional racism is also embedded within the education curriculum through the medical frameworks employed to train medical practitioners. This is either through pathologisation of Aboriginality or through colour blind medical approaches. Colour-blind approaches ignore the extent to which race and racism play a role in people’s health and health-seeking behaviours. Despite the importance of race in examining health disparities in Australia, epidemiological discourses often focus on how to “to close the gap” while conveniently ignoring the processes of racialisation that produce “the gap” [8]. As Gatwiri and Anderson [6] (p.10) argue “continually highlighting health inequalities facing Black people in comparison to white people without a critical theorization of historical, social, cultural, and political contexts, perpetuates the dominant narrative that pathologizes blackness” [20]. The requirement that Aboriginal and Torres Strait Islander Peoples need to be “resilient” in the face of racial discrimination not only undermines the insidious power of racism, it takes the responsibility and impetus for change away from the perpetrators and puts it squarely on those most traumatised and affected by it. 

Embedding critical race theory in health practice offers an opportunity to decolonise the curricula that trains doctors, nurses, and other health professionals and can help reduce the cultural ignorance that is ingrained in medical practices [43]. As such, health inequities between Aboriginal and Torres Strait Islander Peoples and white people in Australia need to be critically considered within the context of the generational trauma of colonisation, assimilation, dispossession, and forced removal from country and community [44].

## 4. Lessons from Yarning with a First Nations Health Researcher

Yarning is an Aboriginal and Torres Strait Islander’s Peoples’ term for talking and storytelling based on reciprocal relationships where respect is assumed” [37]. In research, yarning is a method of data collection characterised by minimal questions and instead requires deep listening (*dadirri*) to allow the stories to unfold and form their own meaning [45]. Barlo and colleagues describe yarning as “a relational methodology for transferring Indigenous knowledge”, and is a flexible process that incorporates Indigenous ways of knowing, being, and doing into research across a broad range of contexts [46]. Yarning circles provide an equal sharing place where equity can be achieved [47], creating a respectful and effective way to prioritise First Nations’ voices within any communication between Indigenous and non-Indigenous group members [48].

A yarning circle among the authors of this paper extended our individual and collective theoretical knowledge shaping our arguments. Darlene Rotumah (2nd author) shared her knowledge as a First Nations health researcher through a yarning circle that was recorded and analysed for content and illustrative quotes within this paper. In our yarning circle, Darlene’s reflections on how Aboriginal Health Workers (AHWs) employed within Australian mainstream health institutions think of their roles was prioritised. In Australia, “Aboriginal Health Workers work within multidisciplinary healthcare teams to achieve better health outcomes for Aboriginal peoples and communities, and play a key role in facilitating relationships between Aboriginal patients and other health professionals” [49]. AHW roles combine professional and cultural obligations, and are often referred to as “cultural workers”. Darlene’s reflections on her research and experiences brings meaning and understanding to AHWs’ cultural obligations and the conflicts of competing priorities in their roles when working within mainstream white-dominated health institutions. Four key learnings emerged from our yarning circle and are discussed below.

### 4.1. Decolonising Healthcare Is Key

The term decolonisation has been used increasingly by academics since the 1980s. It is now vital that it be recognised and demystified within mainstream health institutions as an everyday practice beyond academic theorising. Decolonisation is the process of reclaiming ways of knowing, being, and doing that were/are considered inferior by colonial processes [43]. Sherwood contends that due to its colonial history, Australia “necessitates a contextualised discourse for re-claiming knowledge informed through a balance of truths and histories. Reflecting upon the cause and effect of past action and its policies” [50]. In the context of healthcare, providers working within mainstream health organisations, thinking, and reflecting beyond the dominance of the western biomedical way is the beginning of decolonising the way services are delivered. Decolonisation requires acknowledging that First Nations Peoples’ ways of knowing, being, and doing healthcare have been historically and institutionally marginalised. Australian healthcare institutions would benefit greatly by incorporating Indigenous health knowledges as part of strategies to ‘closing the gap’ [51].

In our yarn, Darlene highlighted that the voices of cultural experts (First Nations Peoples) are often ignored and excluded in medical practice. There are tensions between AHWs who prioritise cultural knowledge in their roles and the practices of Eurocentrism that dominate western medicine. Even though Aboriginal health Workers (AHW’s) culturally specific roles are meant to bridge the existing cultural gap in mainstream health services, they experience repeated disrespect and devaluation of their culturally identified roles from white staff. To counter these racial aggressions in the workplace, AHWs frequently use decolonising healing strategies, such as connection to country to enable them to move past institutional disrespect of their roles. This, however, is often interpreted as neglect of their roles and positioned as incompetence. Darlene reflected:

AHW’s instinctively draw on their cultural knowledge and will take their clients out on country to sit by a river or walk while yarning about their health issues. These relational practices are interpreted as ‘gone walkabout’, or not ‘real’ work.(Darlene, Our Yarn)

Incorporating the core principles of decolonisation by creating culturally safe practices is an effective starting point and framework to assist healthcare workers to decolonise their practice [52]. Practitioners beginning to critically reflect on their unconscious biases, assumptions, and power imbalances and those who adopt cultural humility [53] toward their healthcare practices, can significantly reduce Aboriginal and Torres Strait Islander Peoples’ fear and avoidance of seeking timely healthcare services [54].

### 4.2. Cultural Safety Can Save Lives

Another major theme that emerged from our yarn was the critical role of cultural safety in informing health-seeking behaviours. Literature shows that “unsafe cultural practice comprises any action, which diminishes or disempowers the cultural identity and well-being of an individual” [55]. Cultural safety on the other hand, is promoted by antiracism, which is “an active and consistent process of change to eliminate individual, institutional, and systemic racism” [55]. The cultural safety framework originated from the work of Irihapeti Ramsden in response to her concerns for the lack of safety for Māori people when accessing healthcare [56]. In 1990, the New Zealand Nursing Council incorporated cultural safety into their curriculum assessment processes [57]. They outlined the fundamentals of cultural safety as:

The effective nursing practice of a person or a family from another culture. (This) is determined by that person or family, (where the nurse practitioner undertakes) a process of reflection on his/her own cultural identity and recognises the impact of their personal culture on their professional practice [57].

In our yarn, Darlene discussed two domains of cultural safety. The first was cultural safety for Aboriginal patients and the second was for Aboriginal Health Workers (AHWs). She reflected:

Aboriginal workers experience the same challenges within mainstream health services as their patients. These are experiences of racism, lack of cultural safety, and disregard for cultural knowledge, which make health centres unsafe for both Aboriginal practitioners and Aboriginal patients.(Darlene, Our Yarn)

The lack of racial and cultural safety informs the choice not to access health care services due to fear of re-experiencing racial trauma in their encounters with prejudiced healthcare practitioners. The lack of trust between Aboriginal and Torres Strait Islander Peoples and medical professionals has a colonial history, which has not been acknowledged or mended to date. Darlene added:

Many of our people, don’t feel safe in hospital. I’ve heard people say they would rather die than go to hospital where they are made sicker by encounters of racism. I hear Aboriginal people say “I don’t trust them (medical professionals) … they might want to hurt me deliberately”. This is because we have a history of genocide. This is our history. Our ancestors are genocide survivors. We need more First Nations workers and racially safe hospitals so that our people can feel safe to seek treatment.(Darlene, Our Yarn)

Similarly, Kathomi (first author), described her own experiences as a psychotherapist who works predominantly by providing culturally safe therapy for Black people and people of colour in Australia. She reflected that:

People will choose to sit with pain and wallow in total wounding and hurt rather than go to a white therapist…. Black people come to me to because they know I have an embodied understanding of what it is to live in a Black body in this country… People have been so dehumanised by culturally unsafe services that they say to me ‘I would rather sit here in my home and hurt than go and see another white professional.(Kathomi, Our Yarn)

We all yarned that our lessons from our professional practice and research have clearly shown that First Nations Peoples in Australia may choose to not seek healthcare because racism within healthcare services is experienced as being worse than their health challenges. This is discussed extensively through our own research and the work of others [9,20,58,59,60,61]. 

Cultural education needs to be embedded within medical practice as it educates healthcare practitioners about the historical and contemporary drivers of the health gap between First Nations and white Australians and can increase respectful and culturally safe practices [62]. Although “cultural competence” trainings are mandated in most human-based practices, Darlene reported that AHWs viewed such cultural training programs as western approaches attempting to respond to cultural issues, which they had little understanding of.

I think it’s become like a tick the box type of thing… They continually develop these programs (cultural safety trainings) from a western lens. Yet they will *never* be actually what is needed. I think if they are Indigenous led with Indigenous input…that would be a better model. But, at the moment, they’re developed by white people with black faces delivering them.(Darlene, Our Yarn)

Critical self and institutional reflection are key principles of cultural safety, “aimed at eliminating racism from the health system” [62]. This process enables an examination of how “power imbalances between racialised people and non-racialised/white people play out in the form of unearned privileges that white people benefit from and racialized people do not” [63]. Cultural safety and anti-racism in healthcare are, therefore, powerful and necessary tools with which to fight on the “battlegrounds” of institutional racism.

### 4.3. White Silencing

White silencing has been occurring in Australia since the arrival of the colonising empire. Their version of “discovery”, “settlement”, and subsequent racial, white protectionist policies and practices became the norm therefore silencing and erasing *what was already here*. White supremacist ideology plays a key role in silencing the realities and lived experience of all those who are racially minoritised by whiteness [64]. We define white silencing as the tactics that white people and institutions engage in to preserve white ways of “knowing” and “being” through overt or covert way of punishing those who speak against the dominance of whiteness. Within healthcare, white silencing can have profoundly negative implications. Staff members who call out racism in healthcare are inevitably “silenced” or scapegoated in the workplace. This silencing is enacted by ignoring or minimising the concerns of racism that have been raised. It can also take the form of denial, gaslighting, or punishing by way of workplace lateral violence such as bullying and even threat of or actual loss of employment [64,65]. White practitioners who actively work as anti-racists within health institutions are also frequently silenced by colleagues and employers for “betraying white solidarity” [66]. Elizabeth (third author) reflected on the isolation she experienced as a white anti-racist nurse.

After I started my work and research into anti-racism in the health system, I noticed a shift in how other staff treated me… when I walked into the tearoom, it would go silent… I was accused of prioritising Aboriginal patients over white patients.(Elizabeth, Our Yarn)

Calling out racism in predominantly white settings can trigger defensive responses. The most common is the vehement denial of racism, typically with the white person centering the conversation around their feelings and positioning themselves as the victim. Weaponising white feelings against Black people and people of colour who speak against racism implies that being accused of racist behaviour is more offensive than the acts of racism perpetrated towards Aboriginal and Torres Strait Islander peoples and other people of colour. These reactions reflect the fragility of whiteness as theorised extensively by Robin Di’Angelo [67,68].

Silencing, however, can still involve speaking. White managers can silence by talking about racism without naming the problem.

Some love to do corporate double speak on racism. They will talk and talk around the big elephant in the room but they will never name it. They will never say: ‘the problem is racism’. Without naming the issue, they are performing a form of white silencing. That means, they are speaking, but they are actually silencing the conversation.(Kathomi, Our Yarn)

Silencing also occurs when workplaces acknowledge that “they need to do better”, but never follow up with tangible strategies or solutions to make their workplaces racially or culturally safer. When institutions position themselves as “perpetual learners” of anti-racism, but are deliberately slow on actioning institutional change, this is a form of silencing. It demonstrates that racial and cultural safety are viewed as “add-ons” rather than central tenets of the organisational fabric. This insidious nature of white silencing has a profound, yet unacknowledged impact, in health care settings.

### 4.4. Relational Accountability

The central nature of relationships within First Nations cultures informs the concept of relational accountability [69]. Developed through the work of First Nations scholars [69,70,71,72], relational accountability signals the importance of researchers being accountable towards the relationships they create during their research with First Nations People. When negotiating a health system dominated by western and white protocols, principles of relational accountability can contribute to reducing health inequities [57,73]. This approach is now integrally linked to decolonising education and health as well as promoting the inclusion of Aboriginal and Torres Strait Islander perspectives [74,75,76]. Relational accountability can help to reduce barriers that the individualised, colonial processes create, leaving room for critical examination and consideration of the dynamics of relationships [71,72]. As Darlene reflects:

Relational accountability thinking can assist health professionals to engage wholly in culturally safe practice that goes beyond the current tick-the-box approach to training. There is potential to improve all people’s experiences with health services by work that aims to build trust through a relational practice that enacts the principles of respect, reciprocity, and responsibility.(Darlene, Our yarn)

The three Rs of relational accountability: Respect, Responsibility, and Reciprocity can guide white healthcare professionals to work in a culturally safer way with First Nations Peoples and their communities. Relational methods of doing healthcare have significant potential in improving healthcare for First Nations Peoples and decolonising policy, practice, and everyday interactions in healthcare settings.

## 5. Conclusions

It is well documented that Aboriginal and Torres Strait Islander Peoples’ health in Australia trails that of white Australians. Despite decades of evidence that racism serves as a significant barrier to accessible healthcare for First Nations Peoples, attempts to address this systemic problem have fallen short. The social determinants of health are particularly poignant given the socio-political-economic history of invasion, colonisation, and the subsequent entrenchment of racist practices in Australian healthcare. Employing a critical race perspective to de-center whiteness as the normalised position of a “good and healthy body”, we have argued that Euro-centric, bio-medical discourses, which dominate healthcare in Australia, systemically erase cultural and historical contexts, which inform health seeking behaviours within First Nations communities.

Unless they are decolonised, current health services are culturally unsafe for Aboriginal and Torres Strait Islander peoples who are forced to negotiate health services where racism is experienced both at the interpersonal and institutional level. The ‘othering’ [77] processes and deficit-based lens of Australia’s political and media [78] landscapes not only normalise this, but firmly embed and protect whiteness in healthcare delivery. We conclude that increasing cultural safety, racial dignity, and respect for and inclusion of Indigenous knowledge systems within mainstream health will have a positive outcome. Currently, the impacts of institutional racism continue to marginalise, silence, and harm First Nations Peoples, and, at times, end their lives.

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
