# Peer review of "BlackLivesMatter in Healthcare: Racism and Implications for Health Inequity among Aboriginal and Torres Strait Islander Peoples in Australia"

_ijerph, 2021, doi:10.3390/ijerph18094399_

Round 1
Reviewer 1 Report
This paper contributes to the important and topical discussions on racism and healthcare. The paper is grounded in the social sciences and associated methodologies and yet accessible to a biomedical audience. The authors introduce 3 interesting theoretical approaches which importantly are embedded in Aboriginal and Torres Strait Islander practices. When considering racism in healthcare, much research in biomedical journals simply show data describing healthcare inequities experienced by minoritized ethnic populations. This paper goes a step further and gives some local examples of how racism is translated into differential experiences which lead to these inequalities. In addition, the authors draw out several suggestions on how to change this e.g. decolonisation, cultural humility, cultural safety and avoiding white silencing. These lessons will be widely applicable across different health systems and contexts.
Line 36 Suggest meningococcal infection and sepsis or meningococcal septicaemia
Line 77-78 "theoretical approaches of yearning, storytelling and relational accountability" - I note these methodological approaches are referenced later in the paper but think they should be referenced at this point also
Suggestion only: I am uncertain if the authors have made an explicit decision regarding whether to avoid capitalising 'white' and 'whiteness'. I realised there are arguments for and against capitalisation. From my perspective, I think not capitalising it fails to draw attention to 'Whiteness' as a social category with inherant benefits (White supremacy) and instead neutralises and normalises it - as the authors quote in line 149, I see it as leaving Whiteness as 'unmarked'. Would a footnote be useful to illustrate this thinking?
Author Response
Thank you to the reviewer for their generous comments.
The suggested edit changes have been made.
Thank you for your suggestion on capitalising whiteness. We have considered your argument keenly. We have decided not to capitalise whiteness arguing that it is already the norm and the default.
Reviewer 2 Report
This is an important and interesting paper that appropriately and cogently describes the importance of decolonisation and de-centring of whiteness within healthcare in Australia to move beyond normalisation, assimilation, marginalisation and systemic racism.
p. 2: The 1967 referendum did not provide voting rights to Indigenous peoples in Australia. The discussion of the different definitions of poverty is very welcome.
p. 7-8: A sophisticated discussion of various forms of white silencing through speaking, emotion, and perpetual learning.
Author Response
Thank you for the generous comments. We have changed our inaccurate reference to the right to vote. Many thanks
Reviewer 3 Report
Dear authors, thank you for writing this piece. As a Canadian scholar, I believe that it is somewhat easier to speak this truth in our context than in yours. The recent death of Joyce Echaquan in Quebec led to national meetings on addressing anti-Indigenous racism in the health care system. We assessed the plans these meetings produced as, at best, a tepid response. https://www.cbc.ca/news/canada/manitoba/opinion-anti-racism-canada-health-care-1.5936802.
Having spent time in Australia (one year, primarily in the NT, and many many visits since, because of joint collaborations), I understand your work to be courageous, necessary, and difficult. In our institution, some non-Indigenous well established scholars are taking on the responsibility to speaking up about their own socialization which resulted in prejudice they still fight against. I am one of them. This is settlers' work, and my responsibility.
You will see minor edits in the draft, attached. My only substantive comment is that the title speaks of institutional racism: to me this means the Depts of Health (Commonwealth, state, territories), and hospitals adopting policies on anti-Indigenous racism and having accountability mechanisms when negligence and prejudice, which may result in death, occurs (as in the case of Naomi Williams). Your paper does well on the clinical-interpersonal level, but underdelivers at the institutional level. I suggests adjusting the title, or framing the argument a bit differently. Whatever you chose is fine by me, this needs to be published.

Author Response
Thank you for acknowledging the courage it takes to write on this topic.
We have addressed your suggestions.
This manuscript is a resubmission of an earlier submission. The following is a list of the peer review reports and author responses from that submission.